# TrkB-Induced Inhibition of R-SMAD/SMAD4 Activation is Essential for TGF-β-Mediated Tumor Suppressor Activity

**DOI:** 10.3390/cancers12041048

**Published:** 2020-04-23

**Authors:** Min Soo Kim, Wook Jin

**Affiliations:** Laboratory of Molecular Disease and Cell Regulation, Department of Biochemistry, School of Medicine, Gachon University, Incheon 21999, Korea

**Keywords:** TrkB, transforming growth factor-β (TGF-β), tumor suppressor activity, tumor progression

## Abstract

TrkB-mediated activation of the IL6/JAK2/STAT3 signaling pathway is associated with the induction of the epithelial–mesenchymal transition (EMT) program and the acquisition of metastatic potential by tumors. Conversely, the transforming of growth factor-β (TGF-β) is implicated in tumor suppression through the canonical SMAD-dependent signaling pathway. Hence, TrkB could play a role in disrupting the potent TGF-β-mediated growth inhibition, a concept that has not been fully explored. Here, we identified TrkB to be a crucial regulator of the TGF-β signaling pathway as it inhibits the TGF-β-mediated tumor suppression and the activation of TrkB kinase. We further show that the interactions between TrkB and SMADs inhibit TGF-β-mediated R-SMAD/SMAD4 complex formation and suppress TGF-β-induced nuclear translocation and target gene expression. Additionally, the knockdown of TrkB restored the tumor inhibitory activity of TGF-β signaling. These observations suggest that interactions between TrkB and SMADs are critical for the inhibition of TGF-β tumor suppressor activity in cancer cells.

## 1. Introduction

Transforming growth factor-β (TGF-β) plays a dual role as a potent growth inhibitor of epithelial and non-epithelial cells, inducing growth arrest and apoptosis, and as a promoter of tumor progression in advanced cancers, inducing epithelial–mesenchymal transition (EMT) [1,2]. The growth-inhibitory responses of TGF-β are mediated through the activation of the canonical SMAD-dependent pathway via the induction and reshuffling of CDK inhibitors (CDKN1A). There is ligand-dependent complex formation between TGF-β and type I, and types II serine/threonine kinase receptors phosphorylate the C-terminal SXS-motifs of the receptor-associated SMAD2 and SMAD3 proteins, leading to their activation and subsequent interaction with SMAD4. Following translocation to the nucleus, SMAD complexes induce expression of CDK inhibitors and repression of c-MYC [1,2]. Additionally, TGF-β promotes tumor progression by inducing the expression of specific EMT-associated transcription factors via both canonical SMAD-dependent and non-canonical SMAD-independent signaling pathways [1,3]. 

Neurotrophins and their receptors play a crucial role in the proper development, migration, maturation, survival, and differentiation of the nervous system [4]. In particular, TrkB, a receptor tyrosine kinase for brain-derived neurotrophic factor (BDNF), is essential for the survival and maintenance of synaptic plasticity of the neurons, with both TrkB and BDNF being widely expressed in the central and peripheral nervous system [5,6]. Moreover, TrkB has been implicated in several physiological processes in both neuronal and non-neuronal diseases. Mutations and aberrant expression of TrkB are associated with depression, and schizophrenia, and could even lead to severe obesity in humans [7,8]. TrkB-expressing neurons in the dorsomedial hypothalamus act as critical regulators of energy expenditure and body weight through appetite suppression and an increase in satiety [9]. Furthermore, the downregulation of TrkB expression is associated with the pathogenesis of Alzheimer’s [10,11], Huntington’s [12,13], and Parkinson’s [14,15] diseases. Conversely, TrkB is aberrantly overexpressed in a variety of human cancers [16,17,18]. TrkB potently regulates oncogenesis through the suppression of anoikis [19] and enhancement of Zab1 expression [20]. Previously, we demonstrated that induction of EMT programs by TrkB results in cancer acquiring metastatic potential by suppressing the SOCS3-mediated degradation of JAK2 that subsequently activates the PI3K/AKT and IL6/JAK2/STAT3 signaling pathways [16]. Additionally, TrkB-mediated PI3K/AKT activation was shown to suppress the expression of Runx3 and Keap1 tumor suppressors that are commonly downregulated in malignant tumors [17]. However, the role of TrkB in inhibiting the tumor suppressor activity of TGF-β has not yet been elucidated.

Here, we show that inhibiting the activation of SMADs through the expression of TrkB regulates the tumor suppressor activity of TGF-β. We further demonstrate that TrkB inhibits the nuclear retention of R-SMAD/SMAD4 complexes through the formation of a TGF-β-mediated TrkB/SMAD complex. 

## 2. Results

### 2.1. TrkB Regulates TGF-β-Mediated Tumor Suppressor Activity

The effect of TrkB expression on TGF-β signaling in various TGF-β-responsive cell lines, including RIE-1 (rat intestinal epithelial), HeLa (cervical cancer), and NMuMG (mouse mammary epithelial) cells, was examined using TGF-β1-responsive reporters (SBE-Luc and 3TP-Luc) to determine the role, if any, of TrkB in TGF-β signaling. An altered response of TGF-β-specific luciferase reporter assay was observed in cells transiently transfected with TrkB relative to the activity of the control cells. TrkB significantly inhibits the TGF-β-induced transcriptional activity of SBE and 3TP in RIE-1, HeLa, and NMuMG cells (Figure 1A,B and Appendix AA). These results suggest that TrkB could regulate the canonical TGF-β-signaling pathway. To comprehensively elucidate the role of TrkB in TGF-β-mediated physiological processes and tumor suppression, we generated RIE-1 and HeLa cells that overexpressed TrkB (Appendix AA). This overexpression of TrkB significantly inhibited TGF-β-induced SMAD2 and SMAD3 phosphorylation compared with that of in the control cells (Figure 1C). Thereafter, we examined whether TrkB regulates the expression of TGF-β1 target genes. Notably, the basal expression levels of p15^Ink4b^, p21^Cip1^, PAI1, and TMEPAI were much higher in the absence of TGF-β1 than in the HeLa-control cells. However, the TGF-β1-mediated expression of TrkB drastically reduced a further increase in the expression of the direct target genes of TGF-β1 in the HeLa control cells (Figure 1D). The effect of TrkB on the transcriptional activity of TGF-β1 was assessed using a TGF-β responsive SBE- and 3TP-Luc reporter in NMuMG, RIE-1, NMuMG-TrkB, and RIE-1-TrkB cells. The SBE- and 3TP-luciferase activity following TGF-β1 treatment was markedly enhanced in control cells as compared with that of in NMuMG-TrkB and RIE-1-TrkB cells (Figure 1E and Appendix AB). The transient and stable overexpression of TrkB dramatically reduced the SMAD3-dependent (CAGA)_12_-Luc response (Appendix AC,D). Additionally, the appearance of TrkB dramatically increased the resistance of RIE-1 cells to TGF-β-mediated growth inhibitory activity (Figure 1F). Moreover, RIE-1-TrkB and HeLa-TrkB cells exhibited an enhanced ability for migration and invasion as opposed to RIE-1 and HeLa cells, whereas cells treated with TGF-β1 showed only a slight decrease (Appendix AB). These results suggest that TrkB overexpression inhibits TGF-β-mediated tumor suppressor activity.

### 2.2. Significance of TrkB Tyrosine Kinase Activation in Inhibiting TGF-β Signaling

Activation of TrkB tyrosine kinase has been previously reported to be a requirement for cell survival, tumor metastasis, and the EMT program through the activation of the PI3K-AKT and JAK2-STAT3 pathways [16,18]. These previous reports led us to speculate that the tyrosine kinase activity of TrkB is required for the inhibition of TGF-β-mediated tumor suppressor activity. The significance of activated TrkB in inhibiting TGF-β signaling was assessed by pharmacologically inhibiting TrkB using K252a and observing the effects thereof on the transcriptional activity of SMAD3-dependent (CAGA)_12_-Luc response. SMAD-dependent transcriptional responses of TGF-β1 were significantly restored in RIE-1-TrkB cells, HeLa-TrkB cells, and MDA-MB-231 and Hs578T-TrkB cells that were transiently transfected with TrkB, relative to the control following treatment with K252a (Figure 2A–D). K252a had no effect on the TrkB-mediated inhibition of SBE-luciferase activity in the absence of TGF-β1 (Appendix AC). Additionally, we generated RIE-1 cells that expressed K588M (TrkB KD), a kinase-inactive point-mutant of TrkB, to determine if the tyrosine kinase activity of TrkB is required to inhibit the tumor suppressor activity of TGF-β1 [18]. The effect of the TrkB kinase-inactive mutant on TGF-β signaling was examined using TGF-β1-responsive reporters. Introduction of TrkB KD rescues the TGF-β1-mediated transcriptional activity of SBE, 3TP, and (CAGA)_12_-Luc response relative to that of RIE-1-TrkB cells (Figure 2E–G). Additionally, TGF-β1 significantly stimulated the endogenous phosphorylation of SMAD2 and SMAD3 in RIE-1-TrkB KD cells relative to that of RIE-1-TrkB cells (Figure 2H). These results demonstrate that the activation of TrkB kinases is required for the suppression of the growth inhibitory properties of TGF-β.

### 2.3. Direct Interaction between TrkB and SMADs Inhibits the TGF-β1 Signaling Pathway

Fusion proteins, TrkC and ETV6-NTRK3, inhibit the tumor suppressor activity of TGF-β signaling through physical interaction between the tyrosine kinase domain of TrkC and TGF-β type II (TβRII) receptors [19,20]. Additionally, TrkB suppresses SMAD2 and SMAD3 activation, as shown in Figure 1C. Based on these results, this TrkB-mediated regulatory event possibly occurs upstream of SMAD2 and SMAD3 phosphorylation. Moreover, TGF-β type I and type II receptors could be transcriptionally or translationally regulated by TrkB to downregulate the TGF-β signaling pathway. To test this hypothesis, the expression levels of TGF-β type I (TβRI) and type II (TβRII) receptors were examined in the presence of TrkB. Interestingly, TrkB failed to alter the expression levels of both TβRI and TβRII receptors (Figure 3A). Hence, we sought to identify an interacting partner that specifically recognizes TrkB in the canonical TGF-β signaling pathway to unravel the mechanism of TrkB-mediated regulation. To address this possibility, we firstly examined the possibility of direct interaction between TrkB and TβRI or TβRII receptors by performing coimmunoprecipitation experiments using 293T cells. As shown in Figure 3B, TrkB fails to bind to both TβRI and TβRII receptors. Thus, the specific effects of TrkB on TGF-β signaling are likely elicited through inhibitory activity downstream of TβRI or TβRII receptors.

c-Myc directly associates with the MH2 domain of SMAD2 and SMAD3 to inhibit the TGF-β-induced transcriptional activity of Sp1 and SMAD/Sp1-dependent transcriptions of the *p15^Ink4B^* gene and eventually promotes cell growth and cancer progression [21]. Moreover, the human T-cell lymphotropic virus type 1 (HTLV-1) oncoprotein, Tax, is resistant to the growth inhibitory activity of TGF-β, as interactions with SMADs inhibit the formation of SMAD3/SMAD4 complexes [22]. Consequently, TrkB could inhibit the TGF-β-mediated tumor suppressor activity through direct interactions with SMAD2, SMAD3, or SMAD4, as exemplified by c-Myc and Tax. We, therefore, examined the possibility of direct interactions between TrkB and SMAD2, SMAD3, and SMAD4 by conducting coimmunoprecipitation experiments using 293T cells. Indeed, TrkB strongly associates with SMAD2, SMAD3, and SMAD4 (Appendix AA). Subsequently, we examined the endogenous interaction between TrkB and SMADs in RIE-1 and RIE-1-TrkB cells. Endogenous TrkB directly interacted with the endogenous SMAD2, SMAD3, and SMAD4 proteins (Figure 3C). Thereafter, we investigated whether TrkB expression was correlated with clinicopathological phenotypes in breast cancer patients. TrkB is capable of binding to SMAD2, SMAD3, and SMAD4 in TrkB-expressing human breast tumors. TrkB expression is significantly increased in tissues obtained from breast cancer patients relative to normal breast tissue, with TrkB strongly associating with SMAD2, SMAD3, and SMAD4 (Figure 3D). Deletion mapping experiments using deletion mutants of SMAD2, SMAD3, and SMAD4 indicated that TrkB directly binds to SMADs via the MH2 domain of SMAD2, SMAD3, and SMAD4 (Appendix AA–C). Additionally, we assessed the effect of pharmacologically inhibiting TrkB using K252a to examine if the activation of TrkB tyrosine kinase is required for interactions with SMADs. The interaction between TrkB and SMADs was significantly reduced in RIE-1-TrkB cells following K252a treatments as compared with that of in untreated RIE-1-TrkB cells. Consequently, TGF-β and K252a treatment resulted in a dramatic increase in the expression levels of TGF-β, and phosphorylation of SMAD2 and SMAD3 (Figure 3E).

Moreover, TrkB activation is inhibited by K252a treatment in Hs578T and MDA-MB-231 cells (Appendix AB). The interaction of TrkB with SMAD2, SMAD3, and SMAD3 was also shown to be significantly decreased in the presence of TrkB KD as opposed to TrkB (Appendix AC). Thus, TrkB contributes to the pathogenesis of human breast cancers by inhibiting the TGF-β-mediated tumor suppressor activity via the formation of TrkB/SMAD complexes.

TGF-β interacts with TGF-β receptors to activate R-SMADs (SMAD2 and SMAD3), which in turn form a heteromeric complex with SMAD4. These complexes are subsequently translocated to the nucleus to regulate the expression of target genes [23]. The results, as mentioned above, showed that TrkB interacts with SMAD2, SMAD3, and SMAD4 through their MH2 domains, which also serves as the interaction site for R-SMAD/SMAD4 interactions. Therefore, we hypothesize that TrkB could modify the TGF-β-induced nuclear translocation of SMAD2 and SMAD3. The location of SMAD2 and SMAD3 in the cytoplasmic and nuclear fractions of RIE-1-TrkB cells block nuclear localization of SMAD2 and SMAD3 in response to TGF-β. TrkB expression also blocks the TGF-β-dependent nuclear translocation of SMAD4 relative to the control cells. Moreover, TrkB overexpression resulted in increased expression levels of cytoplasmic SMAD2, SMAD3, and SMAD4 relative to that of control cells. Conversely, TrkB overexpression suppressed the nuclear localization of SMAD2, SMAD3, and SMAD4 relative to that of control cells. The opposite response was observed in TrkB knockdown cells, relative to that of TrkB overexpression (Figure 4A,B and Appendix A). Hence, the role of TrkB in blocking the interactions of SMAD4 with either SMAD2 or SMAD3 was examined to determine the mechanism by which TrkB suppresses the nuclear retention of SMAD2, SMAD3, and SMAD4. We observed a dramatic decrease in SMAD2/SMAD3 and SMAD3/SMAD4 complexes (Figure 4C). These results demonstrate that TrkB inhibits the formation of R-SMAD/SMAD4 complexes, thereby suppressing their translocation and ultimately downregulating the expression of TGF-β1 target genes.

### 2.4. TrkB Knockdown Restored TGF-β Signaling

The highest expression of TrkB was observed in Hs578T and MDA-MB-231 breast cancer cells among the breast cancer cell lines [17]. Hence, we speculated that the loss of TrkB might restore TGF-β-mediated tumor suppressor activity in highly metastatic breast cancer cells. To verify our hypothesis, we used highly metastatic human breast cancer cells (Hs578T and MDA-MB-231) in which endogenous TrkB expression was depleted by short hairpin RNA (shRNA) using a lentivirus system [16] to explore whether loss of TrkB modulates TGF-β signaling in highly metastatic breast cancer cells (Appendix AA). Interestingly, TGF-β signaling was enhanced upon the depletion of TrkB.

The TGF-β1-induced SBE and 3TP transcriptional activity were markedly increased in Hs578T and MDA-MB-231 TrkB-shRNA cells that in Hs578T and MDA-MB-231 control-shRNA cells (Figure 5A,B). Moreover, knockdown of TrkB expression dramatically induced the SMAD3-dependent (CAGA)_12_-Luc response compared with that of the control-shRNA cells (Appendix AA).

These results led us to investigate whether the activation of SMAD2 and SMAD3 could serve as indicators for the activation of TGF-β signaling. TGF-β1 treatment of Hs578T and MDA-MB-231 TrkB-shRNA cells increased the level of SMAD2 and SMAD3 phosphorylation relative to the control-shRNA cells (Figure 5C). Furthermore, consistent with the activation of SMAD2 and SMAD3 in Hs578T and MDA-MB-231 TrkB-shRNA cells, TGF-β1-induced transcriptional regulation of direct target genes was rescued, and TGF-β-mediated growth inhibitory activity was diminished by the loss of TrkB in Hs578T and MDA-MB-231 cells (Figure 5D,E). Moreover, cell proliferation was significantly induced by TrkB relative to that by TrkB KD. However, TGF-β treatment was able to slightly suppress TrkB- and TrkB KD-mediated cell proliferation (Appendix AB). Furthermore, the reintroduction of TrkB in Hs578T and MDA-MB-231 TrkB-shRNA cells drastically inhibited TGF-β-induced CAGA luciferase activity (Figure 5F,G). These results demonstrate that the expression of TrkB in cancer cells requires the suppression of TGF-β-mediated tumor inhibitory activity.

## 3. Discussion

One of the hallmarks of the TGF-β signaling pathway is its pleiotropic nature. Increasing evidence suggests that the oncogenic functions of TGF-β are mediated through their acquired ability to induce EMT and inhibit apoptosis. However, TGF-β can also serve as a tumor suppressor by inhibiting cell proliferation. This is typically mediated by the induction of CDK inhibitors and suppression of c-Myc expression through activation of SMAD-dependent pathways [24].

We recently reported that TrkB induces the EMT program by inhibiting the SOCS3-mediated JAK2 degradation and activating the JAK2/STAT3 pathway. Moreover, TrkB induces the activation of PI3K/AKT and JAK2/STAT3 pathways through activation of c-Src via the formation of TrkB-c-Src complexes [16]. Activation of the TrkB-mediated PI3K/AKT pathway decreases the expression of Runx3 and Keap1 tumor suppressors [17]. However, the mechanism underlying this correlation between TrkB and TGF-β signaling remains unclear. Hence, we extended the study to investigate the function of TrkB in cancer cells and showed that TrkB expression is required for the inhibition of TGF-β signaling to induce the progressive tumor properties of cancer cells.

Our study reveals that TrkB plays an unexpected role in TGF-β signaling. TrkB expression diminishes the antitumoral effects of TGF-β through inhibition of TGF-β signaling pathways. TrkB expression inhibits TGF-β induced SBE4, 3TP, and SMAD3-dependent (CAGA)_12_-luciferase reporter activity, indicating that TrkB affects specific components of the TGF-β mediated signal transduction cascade. TrkB expression also inhibits the TGF-β mediated SMAD2 and SMAD3 activation. These results suggest that TrkB is involved in the inhibition of SMAD-dependent tumor suppressor activity of TGF-β.

Additionally, we previously demonstrated that TrkB expression induces the expression of c-Src, which in turn induces the expression of cyclin D1 [17]. Our results suggest that TrkB inhibits TGF-β-mediated tumor suppressor activity by suppressing the expression of CDK inhibitors as TrkB expression markedly reduced the expression of TGF-β-induced CDK inhibitors. TGF-β is known to induce the expression of the CDK, including p15^Ink4b^ and p21^Cip1^. CDK4- and CDK6-associated p15^Ink4b^ and CDK2-associated p21^Cip1^ inhibit the formation of cyclin D complexes [25] and the formation of cyclin E or cyclin A complexes [26] respectively. These observations are conspicuously consistent with our previous and current results in which TrkB-mediated suppression of the TGF-β1-induced expression of CDK inhibitors promotes the participation of tumor cells in tumor progression.

The TGF-β1-mediated R-SMAD/SMAD4 complex formation is essential to the induction of CDK inhibitors. For transcriptional activation, the SMAD3/SMAD4 complex associated with FoxO transcription factors prior to binding the promoter’s region of the *CDKN2B* and *CDKN1A* genes that encode for p15^Ink4b^ and p21^Cip1^, respectively [27]. Interestingly, TrkB directly associates with SMAD2, SMAD3, and SMAD4 in breast cancer tissues and cancer cell lines. Our data further revealed that interactions between TrkB and the MH2 domains of SMADs inhibit the TGF-β-mediated formation of R-SMAD/SMAD4 complexes as the MH2 domain serves as the binding site for interactions with receptors, partner SMADs, transcriptional coactivators, and chromatin modifiers [28]. Additionally, the lack of interaction between the MH2 domains of SMADs and nucleoporin in the presence of TrkB affects nucleocytoplasmic shuttling, and thus, TrkB also inhibits the accumulation of R-SMAD/SMAD4 complexes in the nucleus of cells [29]. These observations are consistent with previous reports in which c-Myc- and Tax-mediated-TGF-β resistance was achieved through the inhibition of R-SMAD/SMAD4 complexes through direct interactions with SMADs [21,22]. Furthermore, the knockdown of TrkB in breast cancer cells promotes TGF-β-mediated transcriptional activation through phosphorylation of SMAD2 and SMAD3 (Figure 5). Hence, the study suggests the role of TrkB in promoting tumor progression through inhibition of the tumor suppressor activity of TGF-β. These results provide insights into a new mechanism for tumor progression that is mediated by TrkB.

## 4. Material and Methods

### 4.1. Cell Lines, Culture Conditions and Chemical Inhibitors

RIE-1, HeLa, NMuMG, 293T, and human breast cancer cells (Hs578T and MDA-MB-231 control-shRNA or TrkB-shRNA) cells were cultured in Dulbecco’s modified Eagle’s medium (DMEM) supplemented with 10% fetal bovine serum (FBS). TGF-β1 was used at the final concentration of 5 ng/mL for the indicated time. The protein kinase inhibitor K252a was obtained from Abcam.

### 4.2. Plasmids and Viral Production

pLKO.1 lentiviral plasmids encoding human control-shRNA or TrkB-shRNAs have been described [16]. The cDNA encoding human TrkB was subcloned into plenti6.3/V5-TOPO vectors (Invitrogen, Carlsbad, CA, USA), and the TrkB K588M mutant using previously described [30] was generated by site-directed mutagenesis with a Site-Directed Mutagenesis Kit (ThermoFisher Scientific, Telangana, India). To create TrkB or TrkB K588M expression, TrkB or TrkB K588M sequences were inserted in pLNCX-neo vectors, and infected in RIE-1, HeLa, and NMuMG cells and selected with G418. Plasmid transfections were carried out using Lipofectamine 2000 (Invitrogen) reagent, according to the manufacturer’s instructions.

### 4.3. Human Breast Tumor Samples

Proteins extracted from human breast normal and tumor samples were obtained from the Gangnam Severance Hospital after approval by the Institutional Review Board and the ethics committee of Gangnam Severance Hospital (IRB approval number: 3-2011-0191) as previously described [17,31].

### 4.4. Antibodies, Western Blotting, Immunoprecipitation, and Immunofluorescence

We performed Western blotting, immunoprecipitation, and immunofluorescence analysis, as previously described [32]. Antibodies obtained from the following sources: Anti-phospho-SMAD2, anti-phospho-SMAD3, anti-SMAD2, anti-SMAD3, anti-SMAD4, TβRI, TβRII, and TrkB were from Abcam. anti-Flag and β-actin were from Sigma-Aldrich; anti-V5 was from Invitrogen; anti-Myc, α-tubulin, and lamin were from Santa Cruz Biotechnology.

### 4.5. Luciferase Reporter Assay

Cells that were 50% confluent in 12-well dishes were transfected with one of the SBE-4-Luc, 3TP- Luc, SMAD3-dependent (CAGA)_12_-Luc reporter plasmids using Lipofectamine 2000 (Invitrogen). The cell extracts were prepared 48 h after transfection, and the luciferase activity was quantified using the Enhanced Luciferase Assay Kit (BD Biosciences, Franklin Lakes, NJ, USA). All experiments were performed in triplicate.

### 4.6. Thymidine Incorporation Assay

Actively growing asynchronous cells were plated in 24-well plates at a density of 0.5 × 10^5^ cells/well in 0.5 mL culture medium. After waiting 4 h for surface attachment, we incubated the cells in the presence or absence of porcine TGF-β1 (5 ng/mL) for 24 h. The cells were then pulse-labeled with 0.5 µCi [3H]-thymidine for 2 h, fixed with 1 mL methanol/acetic acid, 3:1 (*vol*/*vol*), for 1 h at 25 °C, washed twice with 2 mL 80% methanol, incubated at 37 °C with 0.2 mg/mL trypsin for 30 min and then solubilized with 0.5 mL 1% sodium dodecyl sulfate (SDS). Incorporated activity levels were measured with a scintillation counter.

### 4.7. RNA Preparation and RT-PCR Analysis

Total RNA was isolated using RNeasy Mini Kits (Qiagen, Maryland for North America), and RT-PCR analysis was performed using a One-Step RT-PCR kit (Qiagen) according to the manufacturer’s instructions. The primer sequences used to amplify the investigated genes listed in Appendix A.

### 4.8. Migration Assay

Cells (1 × 10^4^) were resuspended in DMEM medium without FBS and added to the top chambers of 24-well transwell plates (Costar, St. Louis, MO, USA, 8-μm pore size), and DMEM complete media was added to the bottom chambers as an attractant. After incubation, non-migrated cells in the top chamber were removed by swiping with Q-tips, and migrated cells in the bottom chamber were fixed with 20% methanol and stained with crystal violet to visualize the cells. The number of migrating cells in the five fields from each well was counted under 20× magnification.

### 4.9. Statistical Analysis

Data expressed as the means ± SEM. Statistical analyses of the data conducted via the Student’s *t*-test (two-tailed) and ANOVA. Differences were considered statistically significant at *p* < 0.005.

## 5. Conclusions

Activation of TrkB promotes the formation of TrkB/SMADs complex by interacting between the TrkB kinase domain and the Smads MH2 domain and then TrkB suppresses the nuclear retention of SMAD2, SMAD3, and SMAD4. Eventually, TrkB suppresses their translocation and ultimately downregulates the expression of TGF-β1 target genes.

## Figures and Tables

**Figure 1 cancers-12-01048-f001:**
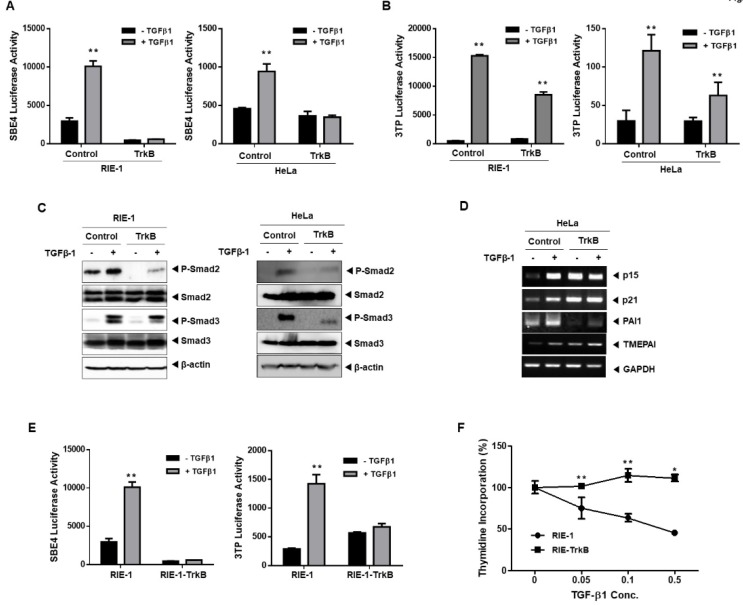
TrkB overexpression inhibits TGF-β-mediated tumor suppressor activity. (**A**,**B**) The activity of TGF-β1-responsive SBE (**A**) or 3TP (**B**) luciferase reporter in RIE-1 and HeLa cells transfected with the TrkB. Luciferase activity was measured 24 h after treatment with TGF-β1 (5 ng/mL). ** Control versus treatment with TGF-β1, *p* < 0.05. *n* = 3. (**C**) Western blot analysis of the expression of phospho-SMAD2, phospho-SMAD3, SMAD2, and SMAD3 in RIE-1, HeLa, RIE-1-TrkB, and HeLa-TrkB cells after stimulation with TGF-β1 (5 ng/mL). (**D**) RT-PCR analyses of p15Ink4b, p21Cip1, PAI1, and TMEPAI mRNAs in HeLa-TrkB cells after treatment with or without TGF-β1 (5 ng/mL). (**E**) Luciferase reporter assay of TGF-β1-responsive SBE or 3TP in RIE-1 or RIE-1-TrkB cells. ** Control versus treatment with TGF-β1, *p* < 0.05. *n* = 3. (**F**) Thymidine incorporation assay of RIE-1 or RIE-1-TrkB cells treated with various concentrations of TGF-β1 as indicated. Points, averages of means from three determinations; bars, SD. * Control versus treatment with TGF-β1, *p* < 0.03. *n* = 3. ** Control versus treatment with TGF-β1, *p* < 0.05. *n* = 3.

**Figure 2 cancers-12-01048-f002:**
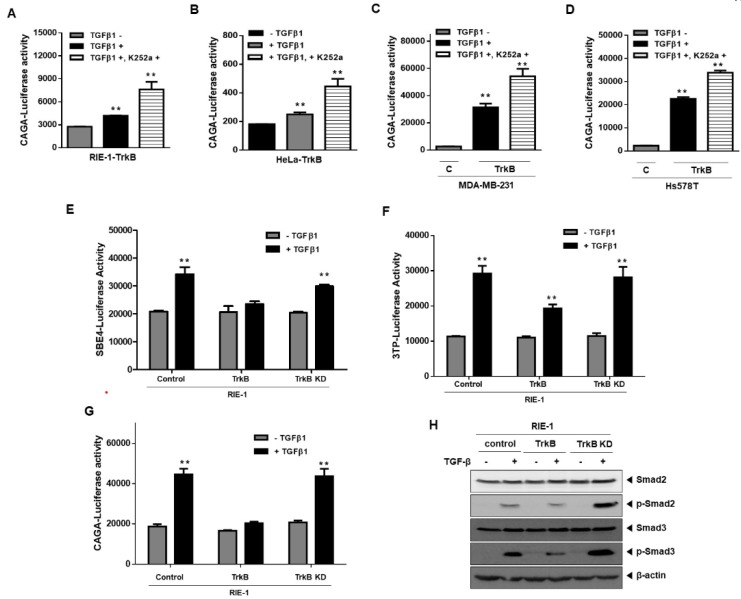
The activation of TrkB kinase required for suppression of the growth inhibitory property of TGF-β. (**A**–**D**) Luciferase reporter assay of SMAD3-dependent (CAGA)_12_-Luc in RIE-1-TrkB cells (**A**), HeLa-TrkB cells (**B**), TrkB-transfected MDA-MB-231 cells (**C**), and TrkB-transfected Hs578T cells (**D**). ** Control versus treatment with TGF-β1, *p* < 0.05. *n* = 3. (**E**,**F**) Luciferase reporter assay of TGF-β1-responsive SBE (**E**) or 3TP (**F**) in RIE-1 cells transfected with the control, TrkB, and TrkB K588M. ** Control versus treatment with TGF-β1, *p* < 0.05. *n* = 3. (**G**) Luciferase reporter assay of SMAD3-dependent (CAGA)_12_-Luc in RIE-1, RIE-1-TrkB, and RIE-1-TrkB K588M cells. ** Control versus treatment with TGF-β1, *p* < 0.05. *n* = 3. (**H**) Western blot analysis of the expression of phospho-SMAD2, phospho-SMAD3, SMAD2, and SMAD3 in RIE-1, RIE-1-TrkB, and RIE-1-TrkB K588M cells after stimulation with TGF-β1 (5 ng/mL).

**Figure 3 cancers-12-01048-f003:**
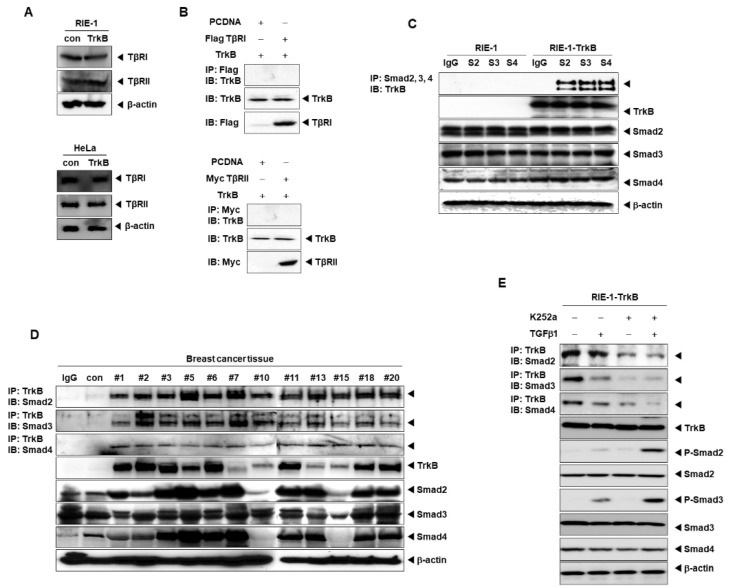
TrkB interacts with SMADs. (**A**) Western blot analyses of the expression of TβRI and TβRII in RIE-1, HeLa, RIE-1-TrkB, and HeLa-TrkB cells. (**B**) Western blot analyses of TrkB following immunoprecipitation of TβRI or TβRII from a whole-cell extract of transfected 293T cells. (**C**) Western blot analyses of TrkB following immunoprecipitation of TβRI or TβRII from a whole-cell extract of RIE-1 and RIE-1-TrkB cells. (**D**) Western blot analyses of TrkB following immunoprecipitation of SMAD2/SMAD3/SMAD4 from a whole-cell extract of tissues of breast cancer patients. (**E**) Western blot analyses of TrkB following immunoprecipitation of SMAD2/SMAD3/SMAD4 from a whole-cell extract of RIE-1-TrkB cells after treated with TGF-β1 (5 ng/mL) or K252a (100 nM).

**Figure 4 cancers-12-01048-f004:**
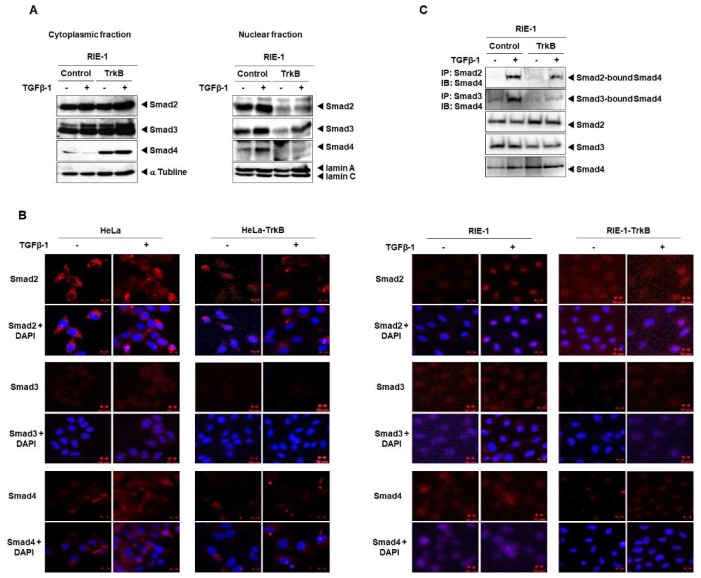
TrkB decreases nuclear retention of SMAD2, SMAD3, and SMAD4. (**A**) TrkB overexpression reduced the level of nuclear SMAD2, SMAD3, and SMAD4. RIE-1, HeLa, and RIE-1-TrkB, and HeLa-TrkB cells treated with TGF-β1 (5 ng/mL) for 1 h. (**B**) Immunofluorescence staining of SMAD2, SMAD3, and SMAD4 in HeLa, RIE-1, HeLa-TrkB, and RIE-1-TrkB cell after treatment of TGF-β1 (5 ng/mL) for 1 h. The scale bar represents 10 μm. (**C**) TrkB inhibits the formation of SMAD2-SMAD4 and SMAD3-SMAD4 complexes. Western blot analyses of SMAD4 following immunoprecipitation of SMAD2 or SMAD3 from a whole-cell extract of RIE-1 control and RIE-1 TrkB cells after treated with TGF-β1 (5 ng/mL) for 1 h.

**Figure 5 cancers-12-01048-f005:**
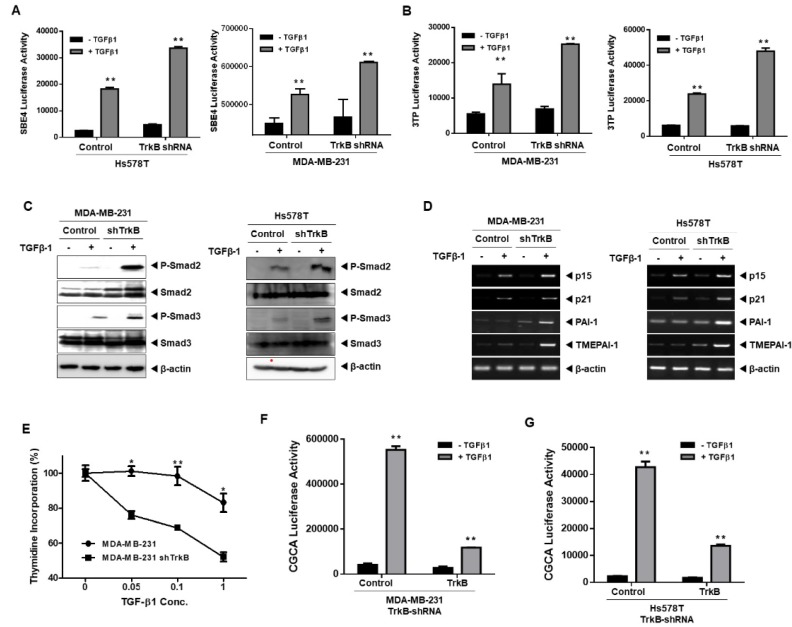
Loss of TrkB restored TGF-β signaling. Luciferase reporter assay of TGF-β1-responsive SBE (**A**) or 3TP (**B**) in Hs578T and MDA-MB-231 control-shRNA or TrkB-shRNA cells. ** Control versus treatment with TGF-β1, *p* < 0.05. *n* = 3. (**C**) Western blot analysis of the expression of phospho-SMAD2, phospho-SMAD3, SMAD2, and SMAD3 in Hs578T and MDA-MB-231-control-shRNA or TrkB-shRNA cells after stimulation with TGF-β1 (5 ng/mL). (**D**) RT-PCR analyses of p15^Ink4b^, p21^Cip1^, PAI1, and TMEPAI mRNAs in Hs578T and MDA-MB-231-control-shRNA or TrkB-shRNA cells after treatment with or without TGF-β1 (5 ng/mL). (**E**) Thymidine incorporation assay of MDA-MB-231 control-shRNA or TrkB-shRNA cells treated with various concentrations of TGF-β1 as indicated. Points, averages of means from three determinations; bars, SD. * Control versus treatment with TGF-β1, *p* < 0.03. *n* = 3. ** Control versus treatment with TGF-β1, *p* < 0.05. *n* = 3. (**F**,**G**) Luciferase reporter assay of SMAD3-dependent (CAGA)_12_-Luc in TrkB transiently transfected MDA-MB-231 (**F**) or Hs578T TrkB-shRNA (**G**) cells. ** Control versus treatment with TGF-β1, *p* < 0.05. *n* = 3.

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
