# Peer review of "TrkB-Induced Inhibition of R-SMAD/SMAD4 Activation is Essential for TGF-β-Mediated Tumor Suppressor Activity"

_cancers, 2020, doi:10.3390/cancers12041048_

Round 1
Reviewer 1 Report
Overall a very interesting paper. Please make the text in the figures more legible. The axes are sometimes hard to read. Also please change 'TGF' to 'TGFb1' in Figure 1A and Figure 5F & 5G. Line 345 is missing a 'Beta'. The largest problem are several small grammatical errors, but the science is interesting and the experiments were carried out well.
Author Response
Overall a very interesting paper. Please make the text in the figures more legible. The axes are sometimes hard to read. Also please change 'TGF' to 'TGFb1' in Figure 1A and Figure 5F & 5G. Line 345 is missing a 'Beta'. The largest problem are several small grammatical errors, but the science is interesting and the experiments were carried out well.
We apologize for this error and have corrected this as suggested by Reviewer #1. Also, this manuscript has been checked by proofreading this manuscript in native English speakers to correct grammatical errors and to improve our manuscript.
Reviewer 2 Report
In this manuscript, the authors carried out a set of experiments that describe the molecular mechanisms by which TrkB inhibits tumor suppressor activity mediated by TGF-b.
Overall, this manuscript is clearly organized and with good hypothesis. The experimental design is straight forward and utilizes appropriate methodologies.
This work is convincing although some points should be considered:
-) The effect of TrkB on the TGF response of 3TP-lux seems to be less significant than the effect on SBE-lux. How do the authors interpret these results ? Could the presence of other response elements than the specific smad elements on the 3TP-lux construct explain this.
-) The level expression of TrkB in cells stably transfected with TrkB should be shown
-) Cell proliferation experiments (thymidine incorporation) are not described in the material and methods section
-) Specificity of action of inhibitor K252a should be shown
-) The effects of K252a in the absence of TGF should also be measured since the overexpression of TrKB affects the activity of SBE-lux EVEN in the absence of TGF-b (Figure 1A)
-) Functionally, is that the overexpression of TrkB KD affects the effects of TGF-b on the proliferation of RIE cells ?
-) Is the mutant form of TrKB (TrKB KD) no longer able to interact with Smad2 or Smad3 ?
Author Response
In this manuscript, the authors carried out a set of experiments that describe the molecular mechanisms by which TrkB inhibits tumor suppressor activity mediated by TGF-b.
Overall, this manuscript is clearly organized and with good hypothesis. The experimental design is straight forward and utilizes appropriate methodologies.
This work is convincing although some points should be considered:
-) The effect of TrkB on the TGF response of 3TP-lux seems to be less significant than the effect on SBE-lux. How do the authors interpret these results ? Could the presence of other response elements than the specific smad elements on the 3TP-lux construct explain this.
We thank Reviewer #2 for this comment. The TGF-β-responsive SBE-luc and 3TP-luc constructs in this manuscript utilized to investigate the regulation of TGF-β signaling by TrkB. The results of SBE-luc and 3TP-luc by TrkB in response to TGF-β may look different in Figures 1A and 1B. However, it has no difference in Figure 1E. Also, (CAGA)12-luciferase reporter plasmid, which is exclusively activated by the TGF-β-induced complex between Smad3 and Smad4, has similar results with results of SBE4-luciferase activity (Figures 2A, 2B, 2C, 2D, 2E, and 2G). Although the results of SBE-luc and 3TP-luc by TrkB in response to TGF-β may look different, TrkB effectively inhibits activation of SBE-luc, 3TP-luc, and CACA-luc activity by TGF-β.
SBE-Luc and CAGA-Luc contain the Smad binding element, but 3TP-Luc contains a TGF-β-mediated plasminogen activator inhibitor-1 promoter. In this view, TrkB mostly inhibits the transcriptional activity of target genes, which containing the Smad binding element, via the suppression of the activation of SMADs than inhibition of TGF-β-mediated plasminogen activator inhibitor-1.
One more thing to think about may be expected to be the result of the difference in transfection efficiency and cell state at the time of the experiment.
-) The level expression of TrkB in cells stably transfected with TrkB should be shown
We confirmed the overexpression of TrkB in RIE-1 and HeLa cells and the knockdown of TrkB in Hs578T and MDA-MB-231 cells by Western blotting and RT-PCR, respectively (Figure S2A).
-) Cell proliferation experiments (thymidine incorporation) are not described in the material and methods section
We have provided the following thymidine incorporation assay in the material and methods section.
Actively growing asynchronous cells were plated in 24-well plates at a density of 0.5 x 105 cells/well in 0.5 ml culture medium. After waiting 4 hr for surface attachment, we incubated the cells in the presence or absence of porcine TGF-b1 (5 ng/ml) for 24 hr. The cells were then pulse-labeled with 0.5 µCi [3H]-thymidine for 2 hr, fixed with 1 ml methanol/acetic acid, 3:1 (vol/vol), for 1 hr at 25°C, washed twice with 2 ml 80% methanol, incubated at 37°C with 0.2 mg/ml trypsin for 30 min and then solubilized with 0.5 ml 1% sodium dodecyl sulfate (SDS). Incorporated activity levels were measured with a scintillation counter.
-) Specificity of action of inhibitor K252a should be shown
K252a known as the TrkB inhibitor, which effectively blocks the kinase activity of Trk. To confirm the specificity of the action of inhibitor K252a, we measured the levels of total and phospho-tyrosine of TrkB in Hs578T and MDA-MB-231 cells with treatment of K252a. There was no change in the protein expression of TrkB, but activation of TrkB significantly inhibited by K252a (Figure S3B).
-) The effects of K252a in the absence of TGF should also be measured since the overexpression of TrkB affects the activity of SBE-lux EVEN in the absence of TGF-b (Figure 1A)
We performed SBE luciferase assay for the effect of K252a in the absence of TGF-β in RIE-1-TrkB and HeLa-TrkB cells, and activation of SBE-luc with K252a treatments has no changed in the absence of TGF-β (Figure S2C).
-) Functionally, is that the overexpression of TrkB KD affects the effects of TGF-b on the proliferation of RIE cells ?
Cell growth assay was measured using RIE-1 TrkB and RIE-1 TrkB KD cells. TrkB significantly induces cell proliferation relative to TrkB KD. However, TGF-β treatment slightly suppresses TrkB and TrkB KD-mediated cell proliferation (Figure S6B).
-) Is the mutant form of TrkB (TrkB KD) no longer able to interact with Smad2 or Smad3 ?
Western blot analysis of V5-TrkB and V5-TrkB KD was performed after immunoprecipitation of the Flag-SMAD2, Flag-SMAD2, and Myc-SMAD4 constructs from whole cell extracts of transfected 293T cells. As a result, the binding of SMAD2, SMAD3 and SMAD4 to TrkB KD was markedly reduced relative to TrkB (Figure S3C).
Reviewer 3 Report
In this manuscript, Kim MS et al show that TrkB inhibits TGF-b pathway by binding with SMAD2/3/4 and reduce P-SMAD2 levels. Data is interesting, but description about the experiments used human breast tumor samples (Figure 3C) are difficult to read.
For better understanding, please change sentence “We next tested the endogenous interaction between TrkB and SMADs in RIE-1 and RIE-1-TrkB cells”(page 6 line192), like “We next tested the endogenous interaction between TrkB and SMADs in RIE-1 and RIE-1-TrkB cells in human breast tumor samples”.
In page 7 line 218, “.. leads to activates ..” may be “.. leads to activate ..”. Please check and correct it. And I suggest to order proofread this manuscript in native English speakers, if possible.
Author Response
In this manuscript, Kim MS et al show that TrkB inhibits TGF-b pathway by binding with SMAD2/3/4 and reduce P-SMAD2 levels. Data is interesting, but description about the experiments used human breast tumor samples (Figure 3C) are difficult to read.
We apologize for this error and have corrected this as suggested by Reviewer #3. Also, this manuscript has been checked by proofreading this manuscript in native English speakers to correct grammatical errors and to improve our manuscript.
For better understanding, please change sentence “We next tested the endogenous interaction between TrkB and SMADs in RIE-1 and RIE-1-TrkB cells”(page 6 line192), like “We next tested the endogenous interaction between TrkB and SMADs in RIE-1 and RIE-1-TrkB cells in human breast tumor samples”
We thank Reviewer #2 for this comment. We changed the sentence as your recommend.
In page 7 line 218, “.. leads to activates ..” may be “.. leads to activate ..”. Please check and correct it. And I suggest to order proofread this manuscript in native English speakers, if possible.
We apologize for this error and have corrected this as suggested by Reviewer #3. Also, this manuscript has been checked by proofreading this manuscript in native English speakers to correct grammatical errors and to improve our manuscript.
Reviewer 4 Report
The paper titled “Inhibited Activation of R-SMADs/SMAD4 by TrkB is Essential to TGF-β-mediated Tumor Suppressor Activity” by Kim and Jin attempts to show that TrkB expression is critical for inhibition of TGF-β signaling in cancer cells. While the paper shows some evidence to support their hypothesis, many of their figures show contradictory data, convoluting their findings. The manuscript also lacks functional studies to show that this pathway can be exploited for cancer therapeutics (e.g. TrkB knockdown in migration and invasion assays). The paper requires extensive revision as the manuscript is poorly written.
- The paper alternates between referring to TrkB and TrkC, two different kinases with varying roles. This occurs in Figure 1 and throughout the manuscript text.
- Some data refute the hypothesis (i.e. overexpression of TrkB enhances TGF-β signaling instead of repressing it). This occurs in Figure 1B, throughout Figure 2.
- Majority of the data uses luciferase assays to show reduction in gene transcription after TrkB overexpression (as seen in Figures 1, 2, and 5), however functional studies (e.g. migration/invasion studies) would have strengthened this report.
- The manuscript requires extensive revision to improve the writing style – many statements throughout the paper are overstated or not comprehensible to the reader.
Author Response
The paper titled “Inhibited Activation of R-SMADs/SMAD4 by TrkB is Essential to TGF-β-mediated Tumor Suppressor Activity” by Kim and Jin attempts to show that TrkB expression is critical for inhibition of TGF-β signaling in cancer cells. While the paper shows some evidence to support their hypothesis, many of their figures show contradictory data, convoluting their findings. The manuscript also lacks functional studies to show that this pathway can be exploited for cancer therapeutics (e.g. TrkB knockdown in migration and invasion assays). The paper requires extensive revision as the manuscript is poorly written.
We apologize for this error and have corrected this as suggested by Reviewer #4. Also, this manuscript has been checked by proofreading this manuscript in native English speakers to correct grammatical errors and to improve our manuscript.
We thank Reviewer #4 for this comment and agree with the Referee’s opinion and suggestion.
As suggested by reviewer #4, we added migration assay using RIE-1, Hela, RIE-1 TrkB, and Hela-TrkB cells in response to TGF-β to support the functional studies. As shown in Figure S2B, TrkB overexpression drastically increases the migration ability of RIE-1 and HeLa cells. The treatment of TGF-β slightly decreases the TrkB-mediated migration ability but not dramatic (Figure S2B).
- The paper alternates between referring to TrkB and TrkC, two different kinases with varying roles. This occurs in Figure 1 and throughout the manuscript text.
We apologize for this error and have corrected this error.
- Some data refute the hypothesis (i.e. overexpression of TrkB enhances TGF-β signaling instead of repressing it). This occurs in Figure 1B, throughout Figure 2.
We appreciate the reviewer’s comments and apologize for these confusions. However, we showed that TrkB expression inhibits TGF-β signaling in all parts of the manuscript, including Figures 1 and 2.
We have corrected our manuscript and also have edited our manuscript by the English language editing service to remove this confusion.
- Majority of the data uses luciferase assays to show reduction in gene transcription after TrkB overexpression (as seen in Figures 1, 2, and 5), however functional studies (e.g. migration/invasion studies) would have strengthened this report.
We appreciate the reviewer’s comments to improve the paper.
As suggested by reviewer #4, we added migration assay using RIE-1, Hela, RIE-1 TrkB, and Hela-TrkB cells in response to TGF-β to support the functional studies. As shown in Figure S2B, TrkB overexpression drastically increases the migration ability of RIE-1 and HeLa cells. The treatment of TGF-β slightly decreases the TrkB-mediated migration ability but not dramatic (Figure S2B).
- The manuscript requires extensive revision to improve the writing style – many statements throughout the paper are overstated or not comprehensible to the reader.
We apologize for this error and have corrected this as suggested by Reviewer #4. Also, this manuscript has been checked by the English language editing service to correct grammatical errors and to improve our manuscript.
Round 2
Reviewer 2 Report
The authors responded to the questions satisfactorily
Author Response
We thank Reviewer #2 for their positive comment and careful review, which helped improve the manuscript.
Reviewer 4 Report
The revised manuscript for “TrkB-Induced Inhibition of R-SMAD/SMAD4 Activation is Essential for TGF-β-Mediated Tumor Suppressor Activity” by Kim and Jin has shown major improvement since initial submission. The manuscript writing has been revised extensively and corrected errors stated in the previous version. However, some data need to be explained.
- TGFβ-induced luciferase activity remains significant when compared to the untreated controls (Figure 1B, 2F), which requires explanation.
- Overexpression of TrkB appears to induce TGFβ target genes (p15, p21, TMEPAI) even without TGFβ1 stimulation, however TGFβ1 stimulation does not further increase their transcription (Figure 1D). This needs to be explained.
- While inhibition of TrkB kinase activity by K252a significantly rescues SMAD3-responsive luciferase activity, TrkB overexpression still induces this response (compared to controls, Figures 2A-2D), which seems to be contradictory to the authors’ hypothesis that TrkB overexpression reduces nuclear translocation of SMAD proteins.
- Statistical analyses are frequently omitted from quantitative figures (Figure 1F, 5E-5G).
Author Response
TGFβ-induced luciferase activity remains significant when compared to the untreated controls (Figure 1B, 2F), which requires explanation.
We thank Reviewer #4 for this comment. The results of SBE-luc and 3TP-luc by TrkB in response to TGF-β may look different in Figures 1A and 1B. However, it has no difference in Figure 1E. Also, (CAGA)12-luciferase reporter plasmid, which is exclusively activated by the TGF-β-induced complex between Smad3 and Smad4, has similar results with results of SBE4-luciferase activity (Figures 2A, 2B, 2C, 2D, 2E, and 2G). Although the results of SBE-luc and 3TP-luc by TrkB in response to TGF-β may look different and activation of 3TP-luc by TrkB still induced by TGF-β, TrkB effectively inhibits activation of SBE-luc, 3TP-luc, and CACA-luc activity by TGF-β (t-test P < 0.05).
SBE-Luc and CAGA-Luc contain the Smad binding element, but 3TP-Luc contains a TGF-β-mediated plasminogen activator inhibitor-1 promoter. In this view, TrkB mostly inhibits the transcriptional activity of target genes, which containing the Smad binding element, via the suppression of the activation of SMADs than inhibition of TGF-β-mediated plasminogen activator inhibitor-1.
One more thing to think about may be expected to be the result of the difference in transfection efficiency and cell state at the time of the experiment.
In addition, We investigate whether the TrkB kinase-dead mutation (TrkB KD) leads to the recovery of TGF-β-mediated inhibition of 3TP-luciferase activity by TrkB. As shown in Figure 2E, and 2F, TrkB expression effectively suppresses SBE-4 and 3TP-luc activity but the introduction of TrkB kinase-dead mutant does not block TGF-β-mediated transcriptional activation of SBE4 and 3TP. These results support that the tyrosine kinase activity of TrkB required for suppressing TGF-β-mediated inhibition of cancer growth.
Overexpression of TrkB appears to induce TGFβ target genes (p15, p21, TMEPAI) even without TGFβ1 stimulation, however TGFβ1 stimulation does not further increase their transcription (Figure 1D). This needs to be explained.
We thank Reviewer #4 for this comment.
TGF-β-mediated cell growth inhibition occurs through induction of cell-cycle-arrest-related genes, such as p15 and p21 and for example, Drak2, as an oncogene, decreases the TGF-β-induced p15, p21, TMEPAI, and PAI1 [1]. Therefore, disruption of TGF-β-mediated growth inhibitory activity caused by elevated levels and activation of oncogenes, such as Drak2 and TrkB, increases their capacity for tumor initiation.
Previously, we showed activation of TrkB tyrosine kinase is required for survival, tumor metastasis, and EMT program via activation of PI3K-AKT and JAK2-STAT3 pathway [2,3].
These our previous results lead to us to speculate that TrkB impart several characteristics to cancer cells, including promoting cell proliferation, elevating ability of migration and motility, increasing disseminating ability, by disruption of TGF-β-mediated growth inhibitory activity.
So, we showed in the manuscript that overexpression of TrkB did not increase p15, p21, and TMEPAI levels by TGF-β (Figure 1D).
However, we have difficulty explaining why the basal level of p15, p21, and TMEPAI without TGF-β increased in HeLa-TrkB cells because we focused on whether TrkB inhibits TGF-β-induced p15, p21, TMEPAI, and PAI1 rather than the basal level of these expressions and we do not have any results.
Nevertheless, we found several clues from previous studies. The abnormal expression of p15, p21, and TEMPAI gene directly participates in the invasion of various human cancers[4-8]. The mutation of p15 gene leads to its abnormal tumor-inducing function, thus facilitating the malignancy and metastasis of tumor cells [4]. Also, p21 highly expressed in AML, glioma, prostate cancer, cervical carcinoma, ovarian cancer, oesophageal SCC, sarcomas, and its upregulation associated with advanced stage and theworst survival of patients[5]. Also, p21 acts as a regulator of TGF-β-mediated tumor growth initiation and local tumor cell invasion in breast cancer[9]. TMEPAI expression was detected mainly in invasive phenotypes of breast cancer cell lines. In addition, the expression of TMEPAI not only promotes growth, migration, invasion, and tumor promoters (HIF-1α and VEGF) but also growth suppression and tumor suppressors (PTEN, TGF-β, and p27kip1) by TGF-β. [7].
So, TrkB may involve in the pathogenesis of cancer through the induction of basal levels of p15, p21, and TMEPAI. However, we do not have any results and clues. Also, in Figure 5D, we can not detect increased basal levels of p15, p21, TMEPAI, and PAI1 in MDA-MB-231 and Hs578T cells control-shRNA cells. So, we think carefully that increased basal levels of p15, p21, TMEPAI, and PAI1 in HeLa cells may occur due to cell-specific phenomenon.
Moreover, if we focus on this phenomenon, it is actually another part of TrkB-mediated regulation of tumor progression.
While inhibition of TrkB kinase activity by K252a significantly rescues SMAD3-responsive luciferase activity, TrkB overexpression still induces this response (compared to controls, Figures 2A-2D), which seems to be contradictory to the authors’ hypothesis that TrkB overexpression reduces nuclear translocation of SMAD proteins.
We thank Reviewer #4 for their positive comment and careful review, which helped improve the manuscript.
K252a is known as the Trk inhibitor, which effectively blocks the kinase activity of Trk. We examined whether suppression of TGF-β-induced CAGA-luciferase activity by TrkB restored by treatment of inhibitor K252a, which is usually used for inhibiting TrkB kinase activation.
As a result, K252a treatment significantly increases CAGA-luciferase activities by TGF-β relative to that of TrkB. We also have similar results (Figure 5A-B). The loss of kinase activity of TrkB by knockdown increases SBE4, and 3TP-luc activation by TGF-β. These results demonstrated that transcriptional activation of target genes, including CDK inhibitors by TGF-β-mediated nuclear translocation of R-SMAD/SMAD4 suppressed by TrkB (Figure 4). We measured the levels of total and phospho-tyrosine of TrkB in Hs578T and MDA-MB-231 cells with the treatment of K252a. There was no change in the expression of TrkB, but activation of TrkB significantly inhibited by K252a (Figure S3B).
Statistical analyses are frequently omitted from quantitative figures (Figure 1F, 5E-5G).
We thank the Reviewer #4 for their positive comment. We apologize for this error and have corrected this as suggested by Reviewer #4.
- Yang, K.M.; Kim, W.; Bae, E.; Gim, J.; Weist, B.M.; Jung, Y.; Hyun, J.S.; Hernandez, J.B.; Leem, S.H.; Park, T., et al. Drak2 participates in a negative feedback loop to control tgf-beta/smads signaling by binding to type i tgf-beta receptor. Cell reports 2012, 2, 1286-1299.
- Kim, M.S.; Lee, W.S.; Jeong, J.; Kim, S.J.; Jin, W. Induction of metastatic potential by trkb via activation of il6/jak2/stat3 and pi3k/akt signaling in breast cancer. Oncotarget 2015, 6, 40158-40171.
- Smit, M.A.; Geiger, T.R.; Song, J.Y.; Gitelman, I.; Peeper, D.S. A twist-snail axis critical for trkb-induced epithelial-mesenchymal transition-like transformation, anoikis resistance, and metastasis. Mol Cell Biol 2009, 29, 3722-3737.
- Yu, C.; Wang, W. Relationship between p15 gene mutation and formation and metastasis of malignant osteosarcoma. Medical science monitor : international medical journal of experimental and clinical research 2016, 22, 656-661.
- Abbas, T.; Dutta, A. P21 in cancer: Intricate networks and multiple activities. Nature reviews. Cancer 2009, 9, 400-414.
- Brunschwig, E.B.; Wilson, K.; Mack, D.; Dawson, D.; Lawrence, E.; Willson, J.K.; Lu, S.; Nosrati, A.; Rerko, R.M.; Swinler, S., et al. Pmepa1, a transforming growth factor-beta-induced marker of terminal colonocyte differentiation whose expression is maintained in primary and metastatic colon cancer. Cancer research 2003, 63, 1568-1575.
- Singha, P.K.; Yeh, I.T.; Venkatachalam, M.A.; Saikumar, P. Transforming growth factor-beta (tgf-beta)-inducible gene tmepai converts tgf-beta from a tumor suppressor to a tumor promoter in breast cancer. Cancer research 2010, 70, 6377-6383.
- Watanabe, Y.; Itoh, S.; Goto, T.; Ohnishi, E.; Inamitsu, M.; Itoh, F.; Satoh, K.; Wiercinska, E.; Yang, W.; Shi, L., et al. Tmepai, a transmembrane tgf-beta-inducible protein, sequesters smad proteins from active participation in tgf-beta signaling. Molecular cell 2010, 37, 123-134.
- Dai, M.; Al-Odaini, A.A.; Fils-Aime, N.; Villatoro, M.A.; Guo, J.; Arakelian, A.; Rabbani, S.A.; Ali, S.; Lebrun, J.J. Cyclin d1 cooperates with p21 to regulate tgfbeta-mediated breast cancer cell migration and tumor local invasion. Breast cancer research : BCR 2013, 15, R49.
